# Current Status of Treatment among Patients with Appendiceal Tumors—Old Challenges and New Solutions?

**DOI:** 10.3390/cancers16050866

**Published:** 2024-02-21

**Authors:** Katarzyna Chawrylak, Magdalena Leśniewska, Katarzyna Mielniczek, Katarzyna Sędłak, Zuzanna Pelc, Sebastian Kobiałka, Timothy M. Pawlik, Wojciech P. Polkowski, Karol Rawicz-Pruszyński

**Affiliations:** 1Department of Surgical Oncology, Medical University of Lublin, Radziwiłłowska 13 St., 20-080 Lublin, Poland; 56646@student.umlub.pl (K.C.); 56506@student.umlub.pl (M.L.); 59465@student.umlub.pl (K.M.); zuzanna.pelc@umlub.pl (Z.P.); sebastian.kobialka@umlub.pl (S.K.); wojciech.polkowski@uml.edu.pl (W.P.P.); karol.rawicz-pruszynski@umlub.pl (K.R.-P.); 2Department of Surgery, The Ohio State University Wexner Medical Center and James Comprehensive Cancer Center, Columbus, OH 43210, USA; tim.pawlik@osumc.edu

**Keywords:** appendiceal tumor, appendiceal malignancy, appendix cancer, appendiceal mucinous neoplasm, gastrointestinal stromal tumor, neuroendocrine tumor

## Abstract

**Simple Summary:**

The 5th edition of the World Health Organization (WHO) classification system for digestive system tumors identifies four types of appendiceal tumors (ATs): serrated lesions and polyps, mucinous neoplasms, adenocarcinomas, and neuroendocrine neoplasms (NENs). Diagnosing ATs poses significant challenges in the medical field due to their uncommon nature and the scarcity of data from large-scale, randomized controlled studies. These tumors are often discovered in tissue samples from appendectomies performed for acute appendicitis. Despite advancements in the treatment of abdominal cancers over recent years, managing and treating ATs effectively remains difficult. This review aims to cover the diagnostic approaches, molecular diagnostics, staging, treatment differences, and prognostic indicators related to ATs.

**Abstract:**

The 5th edition of the World Health Organization (WHO) classification of tumors of the digestive system distinguishes four categories of appendiceal tumors (ATs): serrated lesions and polyps, mucinous neoplasms, adenocarcinomas, and neuroendocrine neoplasms (NENs). The differential diagnosis of ATs can be challenging in medical practice, due to their rarity and lack of data from randomized controlled trials on a large, diverse group of patients. ATs are usually noted in specimens obtained during appendectomies due to clinical acute appendicitis. In the European population, most ATs (65%) occur over the age of 50 and among women (56.8%). According to histological type, 54.6% are neuroendocrine tumors (NETs); 26.8% cystic, mucinous, and serous neoplasms; and 18.6% adenocarcinoma not otherwise specified (NOS). On pathologic analysis, most AT findings are benign lesions or small NENs that do not require further therapeutic measures. The presence of appendiceal mucinous neoplasm (AMN) can lead to pseudomyxoma peritonei (PMP). While the multimodal treatment for abdominal malignancies has evolved over the past several decades, the clinical workup and treatment of ATs remain a challenge. Therefore, this review aims to describe the diagnostic possibilities, molecular-based diagnosis, staging, differences in the treatment process, and prognostic factors associated with ATs.

## 1. Introduction

The 5th edition of the World Health Organization (WHO) classification of tumors of the digestive system distinguishes four large categories of appendiceal tumors (ATs): serrated lesions and polyps, mucinous neoplasms, adenocarcinomas, and neuroendocrine neoplasms (NENs) [1,2]. However, the differential diagnosis of ATs can still be challenging in medical practice, due to their rarity and lack of data from randomized controlled trials involving large, diverse groups of patients [1]. ATs are usually noted in specimens obtained following appendectomies performed due to clinical acute appendicitis [3]. In the European population, most ATs (65%) occur over the age of 50 among women (56.8%). According to histological type, 54.6% are neuroendocrine tumors (NETs); 26.8% cystic, mucinous, and serous neoplasms; and 18.6% adenocarcinoma not otherwise specified (NOS) [4].

The incidence of appendix cancer of primary origin (PAC) is estimated to be 1.2–1.63 per 100,000 people in the United States [5,6]. The 5-year overall survival (OS) depends on the histologic type of the PAC. According to data from a study by Wang et al., 5-year survival is 90.8%, 92.9%, 84.6%, 63.2%, and 96.6% for colonic-type adenocarcinoma, mucinous adenocarcinoma (MAC), NET, signet ring cell carcinoma (SRCC), and malignant carcinoid, respectively [6]. Other variables that influence the 5-year OS are age at diagnosis, race of the patients, tumor size, grading, staging, and administration of chemotherapy (CTH).

The most common clinical manifestation of ATs is acute appendicitis: non-specific abdominal pain, right lower quadrant pain, weight loss, anorexia, fever, vomiting, features of intestinal obstruction from intussusception, and fatigue [7]. In clinical practice, these symptoms usually require appendectomy [8].

On pathologic analysis, most findings are benign lesions or small NENs that do not require further therapeutic measures. Among the benign masses noted on pathology following appendectomy, there are several different types of serrated lesions of the appendix, which are classified as hyperplastic polyps, sessile serrated lesions with dysplasia, and sessile serrated lesions without dysplasia [1,2]. Polyp-like growth can be caused by inflammatory changes in the appendix. The main diagnostic challenge is to distinguish hyperplastic lesions from reactive proliferation. However, if lymphoma or a larger NEN is identified, repeat surgery to obtain a radical surgical margin, cytoreductive surgery (CRS), or CTH might be required for more definitive treatment. In the case of appendiceal adenocarcinoma and mucinous tumors, the lack of data from available studies has resulted in the lack of information to guide specific treatment algorithms [9].

The presence of appendiceal mucinous neoplasm (AMN) can lead to pseudomyxoma peritonei (PMP) [10]. PMP is characterized by peritoneal carcinomatosis, which involves diffusion of intra-abdominal ascites on the surface of the peritoneum. PMP can arise from various neoplasms, including AMN, ovarian tumors, and intraductal papillary neoplasm of the bile duct [11,12,13]. Co-occurrence of PMP with PAC is considered a poor prognostic factor, due to its extensive character, high incidence of recurrence post-surgery, limited treatment efficacy, and the high risk of complications [10].

While multimodal treatment modalities for abdominal malignancies have evolved over the past decades, the clinical workup and treatment of ATs remain a challenge [14]. Therefore, this review aims to describe updated diagnostic possibilities, molecular-based diagnosis, staging, differences in the treatment process, and prognostic factors of ATs.

## 2. Search Strategy

A literature search was performed in PubMed and Google Scholar using the search terms “appendiceal tumor”, “appendiceal malignancy”, “appendix cancer” “appendiceal cancer”, “pseudomyxoma peritonei”, “appendiceal mucinous neoplasm”, “goblet cell carcinoma”, “gastrointestinal stromal tumor of appendix”, and “neuroendocrine neoplasm of appendix”, including articles published between 2000–2023. Filters were used to select clinical trials, meta-analyses, and systematic reviews. Articles published in the past 10 years were prioritized for selection. Frequently cited publications published earlier than a decade ago were also included. The exclusion criteria encompassed publications in languages other than English and book chapters. Information about diagnosis and treatment is mainly based on current guidelines, supplemented with commentary from review articles and studies from different centers to enable the reader to compare individual perspectives and clinical practices.

## 3. Genetics

PAC patient survival, the metastatic burden, and the response to CTH have been associated with tumor molecular subtypes [15]. Ang et al. conducted a study on tumor specimens obtained from 703 cases of appendiceal neoplasms [16]. The specimens were either primary tumors (17.5%), intraperitoneal metastasis (75.0%), or distant metastasis (7.5%). The study noted that appendiceal neoplasms have different molecular profiles than colorectal cancer (CRC). In particular, all appendiceal subtypes had significantly less frequent alterations in *TP53* and *APC* relative to CRC (χ2 *p* < 0.001). Mutation frequencies (%) in five major histopathologic subtypes were noted: in MAC, 77% *KRAS*, 52% *GNAS*, 33% *TP53*, 23% *SMAC4*, 8% *ARID1A*, 6% *APC*, 3% *ERBB2*, 2% *RB1*, and 1.4% microsatellite instability-high (MSI-H); in adenocarcinoma (Ad), 56% *KRAS*, 47% *TP53*, 25% *GNAS*, 18% *SMAC4*, 17% *APC*, 11% *ARID1A*, 3.2% *MSI-H*, 3% *RB1*, and 3% *ERBB2*; in PMP, 81% *KRAS*, 72% *GNAS*, 11% *SMAC4*, 7% *TP53*, 6% *ARID1A*, 4% *ERBB2*, 2% *APC*, 0% *RB1*, and 0.0% MSI-H; in SRCC, 43% *TP53*, 35% *KRAS*, 30% *SMAC4*, 11% *APC*, 8% *GNAS*, 5% *ERBB2*, 3% *ARID1A*, 3.2% MSI-H, and 0% *RB1*; and in goblet cell carcinoma (GCC), 33% *TP53*, 19% *SMAC4*, 15% *ARID1A*, 13% *KRAS*, 6% *GNAS*, 4% *RB1*, 2% *APC*, 2% *ERBB2,* and 2.8% MSI-H. 

*KRAS* mutations were predominantly observed at codon 12, while *GNAS* mutations were localized to codon 201, indicating a prevalent gain-of-function pattern. Conversely, *TP53* mutations exhibited a dispersed distribution across the gene, encompassing numerous frameshift mutations consistent with loss of function. Given the distinctive mutation profile of GCCs compared to other histological types, they were excluded from co-mutation and mutual exclusivity analyses. Significant co-mutation was observed exclusively between *GNAS* and *KRAS*, while mutual exclusivity was identified solely between *GNAS* and *TP53*. Genetic aberrations were subsequently categorized based on signaling pathways. The *RAS/RAF* signaling pathway, encompassing *BRAF, HRAS, KRAS*, and *NRAS*, exhibited the highest frequency of alterations in epithelial appendix cancers, with over 80% prevalence in MACs and PMP, 60% in Ads, and only 33% in GCCs. Alterations in homologous recombination deficiency genes were identified in over 50% of all subtypes, with the highest prevalence observed in SRCC at 80%. These findings underscore substantial molecular distinctions among various appendix cancer subtypes and highlight the prognostic significance of *GNAS* and *TP53* mutation statuses.

Performing next-generation sequencing (NGS) helps to establish a diagnosis in cases in which it is uncertain whether the low-grade mucinous tumor originates from the ovary or from the appendix. KRAS and GNAS co-mutations favor a diagnosis of low-grade appendiceal mucinous neoplasm (LAMN) [17].

Considering AMN, both LAMNs, high-grade appendiceal mucinous neoplasm (HAMN) and MAC, have mutations of oncogenes, including *KRAS, GNAS, TP53*, and *RNF43*, with similar rates. However, patterns of mutations are different in each subtype of AMN. There is also a possibility that LAMN may gradually transform into MAC via HAMN; however, more research on this topic is needed [18].

Foote et al. identified distinct molecular subtypes: a clinically indolent RAS-mut/GNAS-wt/TP53-wt subtype, a CTH-resistant GNAS-mut predominant subtype, and an aggressive, highly aneuploid TP53-mut predominant subtype. Each subtype exhibited specific clinical behavior regardless of its histopathological type [15].

## 4. Grading and Staging of ATs

Currently used staging protocols for appendix neoplasms are based on the 8th edition of the American Joint Commission on Cancer (AJCC) staging system; however, the 9th edition was recently published [19,20]. The staging system for carcinomas of the appendix (Table 1 and Table 2) applies to adenocarcinomas (and variants), GCC, mucinous neoplasms, and small-cell and large-cell (poorly differentiated) neuroendocrine carcinoma [21,22]. The 8th AJCC protocol for NET (Table 3) of the appendix is applicable to well-differentiated neuroendocrine tumors. The ENET guidelines can be used interchangeably in the assessment of NET staging [23]. There is an 8th AJCC protocol exclusively for GIST (Table 4).

The heterogeneous nature of ATs poses a diagnostic challenge and results in difficulties in establishing universal guidelines [20].

### 4.1. Mucinous Neoplasms

The International Agency for Research on Cancer (IARC)/WHO, 2019, distinguishes the following types of AMN: serrated lesions with or without dysplasia and hyperplastic polyps, LAMN, HAMN, and MAC [1].

Serrated lesions and polyps have a serrated architecture of the crypt lumen with or without atypia and intact muscularis mucosae. LAMN has specific histologic criteria: low-grade cytology, rare, not atypical mitotic activity, filiform villi, undulating, flat architecture, pushing type of invasion [21], dissection of acellular mucin in the wall, and fibrosis of the submucosa [24].

Focal obliteration or loss of the lamina propria and muscularis mucosa are the minimum requirements for pTis LAMN. Acellular mucin does not extend through the subserosa [24].

Two categories of LAMN are distinguished: type I, in which the tumor remains confined to the lumen of the appendix, and type II, characterized by the presence of mucin and/or the tumor epithelium extending into the appendix submucosa, wall, and/or surrounding tissues, potentially causing perforation [25]. When acellular mucin or the mucinous epithelium are noted in the subserosa or mesoappendix but do not extend to the visceral peritoneal surface, the tumor should be classified as pT3 [21,24]. Invasion to the visceral peritoneum, including focal mucin or the mucinous epithelium involving the serosal surface or mesoappendix, should be designated as pT4a. When the tumor directly invades or adheres to adjacent structures and organs, it is classified as pT4b [24].

The histologic features distinguishing HAMN from LAMN are high-grade cytology with marked atypia and frequent, usually atypical mitotic activity [21].

MAC has an infiltrative type of invasion, which includes infiltrative glands, incomplete glands, or single infiltrative tumor cells associated with extracellular mucin and desmoplastic stroma. A second pattern of infiltrative invasion is a “small cellular mucin pool” characterized by small dissecting pools of mucin containing floating nests, glands, or single neoplastic cells.

A distinction is made between MAC with signet ring cells when the signet ring cell component accounts for equal to or less than 50% of the tumor cells and mucinous SRCC when the signet ring cell component accounts for more than 50%. Grading of AMN according to WHO is as follows: LAMN-G1, HAMN-G2, MAC-G2, and MAC with signet ring cells-G3 [21].

The dissemination of AMN in the peritoneum is associated with a specific clinical condition, such as PMP. Dissemination may be discovered during an appendectomy or be the result of recurrence. PMP is characterized by mucosal deposits implanted in the visceral peritoneum, on the intraperitoneal organs, and in the abdominal cavity and pelvis. The appendix is not the only origin of PMP. Other reported primary origins are ovarian, colonic, and pancreatic [10,23,26].

PMP can be classified into two distinct groups, namely unique methylation epigenotype (UME) and normal-like methylation epigenotype (NLME), determined by the methylation patterns in their promoters. The involvement of genes linked to neuronal development and synaptic signaling may contribute to the development of PMP [27]. The most frequent genetics aberrations in PMP are similar to those in AMN and include *KRAS, GNAS*, *TP53, ATM, ERBB2, FBXW7, NRAS*, and *SMAD4* mutations [27,28]. In addition, PMP can present with heterogenic histologic features, for example, an epithelium with both low-grade and high-grade cytologic atypia can be found in the specimens. Moreover, the specimens may be rich and poor cellular and may or may not contain signet ring cells. Based on these features, tumor grading can be performed [21,28].

### 4.2. Neuroendocrine Neoplasm

NEN is a well-differentiated appendiceal epithelial neoplasm and is usually associated with a good prognosis and an almost 100% 5-year OS. Appendiceal NEN has been reported in adult patients as well as pediatric patients, most commonly found at the tip of the appendix [23,29]. It is usually found accidentally, causing the symptoms of appendicitis. The occurrence of tumors leading to carcinoid syndrome is rare and is associated with metastases. Immunohistochemical staining for synaptophysin and chromogranin A is needed to confirm the neuroendocrine origin of NEN [30]. Both the mitotic count per 10 HPF and the Ki-67 index should also be reported because the WHO grading classification of NEN is based on these factors [23].

### 4.3. Gastrointestinal Stromal Tumors (GISTs) of the Appendix

GISTs are mesenchymal tumors, rarely found among appendix specimens, and most stain positive for CD117 (C-Kit), CD34, and/or DOG-1 [31]. Almost all appendiceal GISTs are indolent; however, aggressive behavior (invasion and perforation of adjacent bowel) has been reported [32].

Hu et al. reported 27 cases of appendiceal GISTs, with a median patient age of 68 years (range: 34–83) and a male–female ratio of 1:2 [33]. In 85% of patients, the tumor was asymptomatic and was found accidentally during endoscopic or imaging examinations or intra-abdominal surgery for another reason. The remaining 15% of patients presented with symptoms of appendicitis. All tumors displayed a spindle-shaped cell structure, and none exhibited any signs of tumor necrosis. Molecular analysis was conducted on two tumors: one revealed a deletion within KIT exon 11, while the other did not exhibit any mutations in KIT, PDGFRA, or NF1. There were no germline mutations found in any specimen. No disease progression was observed in any of the patients after appendectomy. However, for better insight into this type of ATs, studies on a larger group are required [33].

### 4.4. GCC

GCC accounts for approximately 15% of PAC [5]. A typical tumor may not form a mass detectable radiologically or by macroscopic examination of the specimen, as it infiltrates the appendix wall diffusely. The clinical picture depends on stage and in the case of low-grade GCC may resemble appendicitis [34]. Acute appendicitis is a rare manifestation of high-grade GCC. Symptoms are more often associated with metastases, which are usually located in the peritoneum [34,35]. Peritoneal dissemination is the most common cause of disease-specific death [36]. The WHO grading system for GCC is based on the proportion of tumors with a low-grade tubular/clustered growth pattern (%), which is >75%, 50–75%, and <50% for G1 (low grade), G2 (intermediate grade), and G3 (high grade), respectively. For stage IV tumors, grading may differ between the primary tumor and peritoneal/ovarian metastases. Following standard practice, the grade of the metastatic lesion should be assigned [37,38].

The histologic criteria defining the classification of GCC involve a distinctive pattern characterized by a classic round, tubular growth structure primarily composed of goblet-like mucinous cells, alongside a lesser proportion of Paneth-like and endocrine-like cells. In low-grade GCC, the presence of extracellular mucin pools containing round tubules or cohesive clusters, including ruptured tubules, is a common feature. High-grade GCC manifests through five predominant patterns: the diffused infiltration of signet ring-like/goblet-like cells either as single cells or in abortive tubules, fusion of clusters leading to the formation of large anastomosing structures or extensive aggregates, infiltrative single-file or anastomosing cords of tumor cells with high-grade nuclei, gland-forming adenocarcinoma often displaying a micro-glandular growth pattern of tubules with high-grade nuclei, and a solid sheet-like growth lacking clustered formations and exhibiting minimal to no intracytoplasmic mucin. High-grade GCC presents with heightened architectural and cytologic atypia compared to low-grade GCC, showing increased mitotic activity, and may evoke a desmoplastic response, along with necrosis [38]. Synaptophysin and chromogranin expression in GCC can be diffuse and strongly positive, but typically, staining is only focal [34]. The prognostic and diagnostic value of the *Ki67* index and immunohistochemical markers, such as *CDX2, SATB2, CK7*, and *CK20*, in GCC may be present, although their utility is constrained [38].

### 4.5. Treatment

The choice of treatment for appendix tumors presents numerous challenges due to their rarity, diagnostic issues, and, consequently, the absence of a gold-standard approach [14].

Conducting a preoperative CT assessment of ATs proves advantageous by providing an approximate evaluation of mass size, identification of potential metastases, and sometimes determination of tumor type. This assessment aids in informed decision making about the most suitable surgical approach for a given case [39].

Endoscopic ablation or minimally invasive appendectomy remains the recommended treatment for serrated lesions of the appendix, due to their benign nature. This procedure is beneficial in the case of lesions found accidentally during endoscopic examination, providing histopathological diagnosis or a less traumatic alternative for the patient in the absence of signs indicating appendicitis. Follow-up should be performed in accordance with the local colon-cancer-screening programs [40,41].

If adequate resection is achieved during appendectomy for LAMN, with no accompanying perforation, PMP, or mucinosis, appendectomy alone is an adequate treatment [42]. Meanwhile, when LAMN occurs in the setting of perforation, dissemination of cellular mucin into the peritoneal cavity, or PMP, studies indicate the advantages of combining appendectomy with CRS and hyperthermic intraperitoneal chemotherapy (HIPEC) [43,44,45]. There are no specific guidelines on a chemotherapy regimen; however, oxaliplatin and mitomycin are the preferred cytostatic agents during HIPEC. However, due to its slightly higher hematologic toxicity and lower impact on the quality of life compared to oxaliplatin in HIPEC, mitomycin might be a preferable option for patients with previous CTH and thrombocytopenia. Conversely, oxaliplatin could be favored in patients experiencing leukopenia [46]. Chen et al. evaluated CRS+HIPEC treatment for PMP coexisting with other types of ATs (not LAMN/HAMN), in combination with neoadjuvant CTH (NAC). NAC was identified as an independent factor associated with a significantly worse OS [47]. A promising avenue for future research in the treatment of PMP seems to involve the refinement and exploration of radical exenteration and transplantation techniques. This is indicated by the observed enhancement in the patient’s quality of life, despite the challenges posed by surgically related complications, leading to a 79% 1-year and a 55% 5-year survival rate. The notable 91% rate of disease progression/recurrence in patients monitored for over 6 months underscores the imperative for additional investigation to optimize long-term efficacy and outcomes [48]. Data are scarce regarding the treatment of HAMN, and urgent clinical practice often mirrors the approach used for individuals with LAMN (Figure 1) [37].

NEN <1 cm is mainly treated surgically and do not require observation. In the case of R1 resection or tumors located at the base of the appendix, right colectomy with lymphadenectomy is recommended [23]. Nest et al. provided evidence that indicated right colectomy is not warranted after complete resection of an appendiceal NET measuring 1–2 cm with an appendectomy. Regional lymph node metastases in appendiceal NETs might not have clinical relevance. Additionally, the necessity for further postoperative exclusion of metastases and histopathological assessment of risk factors has been questioned [49]. Tumor size >2 cm is an indication for right colectomy. Follow-up should be provided in each case of NEN >1 cm [50] (Figure 2).

There is a lack of high-quality randomized controlled trial evidence to support any specific treatment strategies for appendiceal GISTs. Nevertheless, in most cases, appendectomy alone appears to yield favorable outcomes [33,51]. In malignant GISTs, CTH (i.e., imatinib mesylate) might be applicable, but research is still ongoing [32].

For patients with GCC tumors categorized as stage pT3 or pT4, right colectomy is associated with improved 5-year survival. However, this benefit is not observed in pT1–2 patients. Adjuvant CTH has been associated with a positive outcome for stage III or higher-grade tumors. In the case of synchronous metastasis, most studies advocate palliative CTH. However, CRS + HIPEC among patients with appendiceal GCC and limited peritoneal spread demonstrate promising potential for long-term survival achievement [52,53].

Future perspectives on the diagnosis and treatment of ATs should focus on systematizing the tools needed for the diagnosis of ATs, which, due to their rarity and nuanced nature, remain a challenge all the time. Treatment of ATs also remains a challenge due to the inability to conduct large, randomized trials. However, with numerous original reports, we should aim to systematize guidelines for all ATs.

## 5. Prognostic Factors

Prognostic factors of ATs vary depending on the histological type and clinical presentation of the disease [54]. Higher-grade tumors often indicate a more aggressive disease and may have a worse prognosis. The extent of peritoneal spread and staging can significantly impact prognosis, while complete cytoreduction with negative margins can positively impact outcome [5,55]. AMN is associated with poor survival, particularly among individuals with a high peritoneal cancer index (PCI) [55,56]. Van Ruth et al. reported a correlation between carcinoembryonic antigen (CEA) and carbohydrate antigen 19-9 (CA 19-9) levels and tumor size in 63 PMP patients. Patients with elevated initial or consistently non-normalizing levels of CA 19-9 exhibited notably higher recurrence [57]. Carmignani et al. conducted a study on 532 patients with PMP. Normal preoperative tumor markers correlated with a significantly improved survival. While elevated CEA during recurrence was associated with significantly diminished prognosis, CA 19-9 levels failed to correlate with outcomes [58]. Kozman et al. identified the CA19-9/PCI ratio as an independent prognostic factor for the OS in patients with low-grade PMP undergoing CRS and integrated palliative and oncology care (IPC). This novel index acted as a surrogate of tumor biology by considering both tumor activity and volume simultaneously, offering a valuable tool to guide treatment decisions in such specific patient cohorts. An elevated CEA/PCI ratio appears to compound the negative effect on the OS [59]. Nummela et al. described tissue expression patterns of CEA using immunohistochemistry in 91 cases of appendiceal PMP. There was a correlation between CEA serum levels and PCI, and no association between biomarkers and histological subtype or prognosis was noted [60]. Wang et al. analyzed survival and prognostic factors for PAC in a cohort of 2891 cases. Patients aged ≥69 had the highest risk of death among all age groups, and the risk of death was higher among Black versus White patients. Poorly differentiated adenocarcinoma and SRCC were negative prognostic indicators. Patients with low-grade tumors had a favorable prognosis. Patients with distant metastases did not experience a prolonged survival benefit from CTH [6]. Fleischmann et al. evaluated the role of regional lymph node (RLN) retrieval on stage migration, OS, and cancer-specific survival (CSS) in appendiceal cancer. The incidence of node-positive cancer was correlated with an increasing number of retrieved RLNs, up to approximately 10 RLNs. However, beyond retrieving 10 RLNs, the incidence of nodal metastasis did not increase further. Retrieving 12 or more RLNs did have a positive benefit on the OS [61]. Similarly, in the study by Raoof et al. that included 573 patients, tumors that were larger than 1 cm had a worse OS when 12 or fewer lymph nodes were identified [62].

## 6. Conclusions

Recent data indicate a rising number of ATs. However, existing research on prognostic factors and treatment algorithms remains inadequate to establish universal guidelines relative to disease management. Distinct staging protocols based on the histological type of cancer hold prognostic value and should aid in selecting tailored procedures for individuals. Completeness of cytoreduction supplemented with HIPEC remains a crucial factor to achieve satisfactory short- and long-term outcomes among patients with ATs.

## Figures and Tables

**Figure 1 cancers-16-00866-f001:**
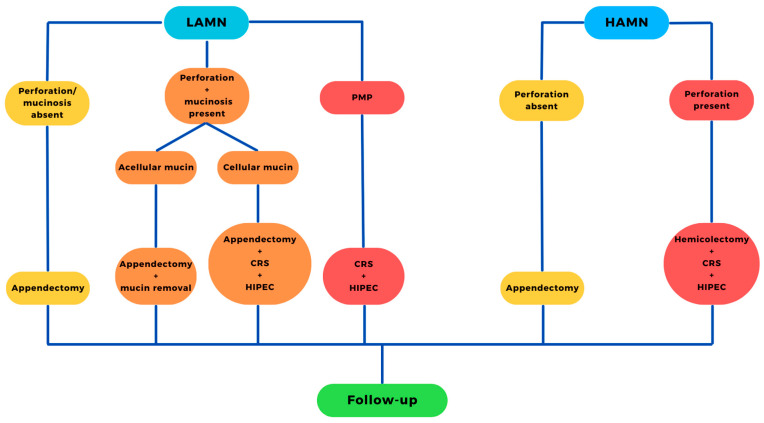
Recommended therapy for AMN. LAMN: low-grade appendiceal mucinous neoplasm; HAMN: high-grade appendiceal mucinous neoplasm; HIPEC: hyperthermic intraperitoneal chemotherapy; CRS: cytoreductive surgery; PMP: pseudomyxoma peritonei.

**Figure 2 cancers-16-00866-f002:**
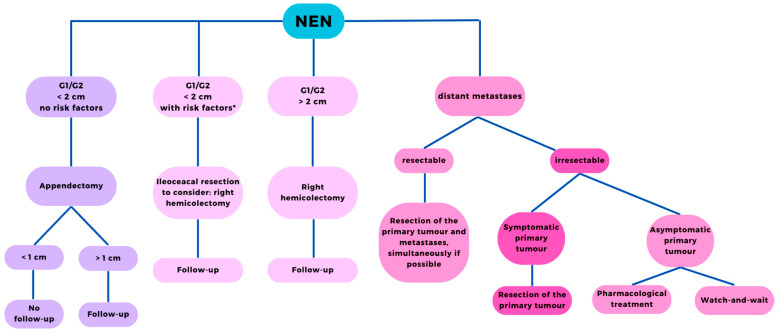
Recommended therapy for NENs of the appendix. * Risk factors: R1 resection, angioinvasion, infiltration of mesoappendix >3 mm, and KI-67 > 2%; NEN: neuroendocrine tumor.

**Table 1 cancers-16-00866-t001:** The 8th edition of the American Joint Commission on Cancer (AJCC) staging system for carcinomas of the appendix (ICD-10: C18.1) [18].

T—primary tumor	
TX	The assessment of the primary tumor is not feasible.
T0	No indication of a primary tumor.
Tis	Carcinoma in situ (intramucosal carcinoma, with the invasion of the lamina propria or extension into the muscularis mucosae, without penetration through it).
Tis (LAMN)	Low-grade appendiceal mucinous neoplasm confined by the muscularis propria. Infiltration of acellular mucin or the mucinous epithelium into the muscularis propria can occur. The T1 and T2 classifications are not applicable to low-grade appendiceal mucinous neoplasm (LAMN). If acellular mucin or the mucinous epithelium extends into the subserosa, it should be categorized as T3, and if it reaches the serosa, it should be classified as T4a.
T1	The tumor infiltrates the submucosa by penetrating through the muscularis mucosa but without extending into the muscularis propria.
T2	The tumor infiltrates the muscularis propria.
T3	The tumor infiltrates through the muscularis propria into the subserosa or the mesoappendix.
T4	The tumor infiltrates the visceral peritoneum, incorporating acellular mucin or the mucinous epithelium that affects the serosa of the appendix or mesoappendix, or directly invades nearby organs or structures.
T4a	The tumor penetrates through the visceral peritoneum, encompassing the presence of acellular mucin or the mucinous epithelium involving the serosa of the appendix or the serosa of the mesoappendix.
T4b	The tumor directly infiltrates or adheres to adjacent organs or structures.
N—regional lymph nodes (including ileocolic nodes)	
Nx	Assessment of regional lymph nodes is not feasible.
N0	There is no evidence of metastasis to regional lymph nodes.
N1	One to three regional lymph nodes exhibit positivity, indicated by the presence of a tumor in the lymph nodes, measuring > 0.2 mm, or the existence of tumor deposit(s) alongside negative lymph nodes.
N1a	A single regional lymph node is positive.
N1b	Two or three regional lymph nodes are positive.
N1c	No regional lymph nodes are positive, but there are tumor deposits in the subserosa or mesentery.
N2	Four or more regional lymph nodes are positive.
M—distant metastasis (in cases where specimens comprise acellular mucin without identifiable tumor cells, if additional tissue is available, it should be submitted to comprehensively assess for the presence of tumor cells.)	
M0	No distant metastasis
M1	Distant metastasis
M1a	Intraperitoneal acellular mucin, without identifiable tumor cells in the disseminated peritoneal mucinous deposits
M1b	Intraperitoneal metastasis only, including peritoneal mucinous deposits containing tumor cells
M1c	Metastasis to sites other than the peritoneum

Applicable to Ads (and variants), GCC, mucinous neoplasms, and small-cell and large-cell (poorly differentiated) neuroendocrine carcinoma.

**Table 2 cancers-16-00866-t002:** Stage grouping for carcinomas of the appendix (ICD-10: C18.1) [21].

Stage 0	Tis	N0	M0	
	Tis (LAMN)	N0	M0	
Stage I	T1-T2	N0	M0	
Stage IIA	T3	N0	M0	
Stage IIB	T4a	N0	M0	
Stage IIC	T4b	N0	M0	
Stage IIIA	T1-T2	N1	M0	
Stage IIIB	T3-T4	N1	M0	
Stage IIIC	Any T	N2	M0	
Stage IVA	Any T	Any N	M1a	
	Any T	Any N	M1b	G1
Stage IVB	Any T	Any N	M1b	G2, G3, GX
Stage IVC	Any T	Any N	M1c	Any G

Applicable for adenocarcinomas (and variants), GCC, mucinous neoplasms, and small-cell and large-cell (poorly differentiated) neuroendocrine carcinoma.

**Table 3 cancers-16-00866-t003:** The 8th edition of the American Joint Commission on Cancer (AJCC) staging system for NET of the appendix [22] and differences in staging: ENET guidelines [24].

The 8th edition of the AJCC staging system for NET of the appendix [22]		Differences in staging: ENET guidelines [24]
T—primary tumor		
T0	There is no indication or evidence of a primary tumor.	
T1	The tumor measures 2 cm or less in its largest dimension.	
T2	The tumor measures more than 2 cm but is less than or equal to 4 cm in its greatest dimension.	The tumor, with a size of ≤2 cm, exhibits infiltration into the submucosa and the muscularis propria and/or minimal (≤3 mm) infiltration of the subserosa and/or mesoappendix.
T3	The tumor measures more than 4 cm or displays subserosal invasion or involvement of the mesoappendix.	The tumor is greater than 2 cm and/or demonstrates extensive (>3 mm) infiltration of the subserosa and/or mesoappendix.
T4	The tumor perforates the peritoneum or directly invades other adjacent organs or structures, excluding direct mural extension to the adjacent subserosa of the adjacent bowel (e.g., abdominal wall and skeletal muscle).	The tumor exhibits infiltration of the peritoneum and/or another adjacent organ.
N		
Nx	Regional lymph nodes cannot be assessed.	
N0	No regional lymph node metastasis	
N1	No regional lymph node metastasis	
M		
M1	Distant metastasis	
M1a	Liver metastasis	
M1b	At least one extrahepatic site of metastasis (e.g., lung, ovary, nonregional lymph node, peritoneum, bone)	
M1c	Both hepatic and extrahepatic metastases	

**Table 4 cancers-16-00866-t004:** The 8th edition of the American Joint Commission on Cancer (AJCC) staging system for GIST of the appendix [22].

T—primary tumor	
Tx	Primary tumor cannot be assessed.
T0	No evidence of a primary tumor
T1	Tumor ≤ 2 cm
T2	Tumor >2 cm but ≤5 cm
T3	Tumor >5 cm but ≤10 cm
T4	Tumor ≤10 cm in the greatest dimension
N	
N0	No regional lymph node metastasis
N1	No regional lymph node metastasis
M	
M1	Distant metastasis

## Data Availability

The data presented in this study are available on request from the corresponding author (accurately indicate status).

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
