# Peer review of "Current Status of Treatment among Patients with Appendiceal Tumors—Old Challenges and New Solutions?"

_cancers, 2024, doi:10.3390/cancers16050866_

Round 1
Reviewer 1 Report
Comments and Suggestions for Authors
This is a narrative review about the treatment of appendiceal tumors.
The topic is interesting and almost completely exhaustive.
A moderate revision of the English language is needed
The concepts reported by the authors are very well known and a section on future perspectives would be useful, including clinical implications of the treatment as well as of the disease
It would also be useful to expand the references by updating them:
- doi: 10.1159/000536219
- https://doi.org/10.1007/978-3-031-36860-8_9
- doi: 10.7759/cureus.50783
Many others
Comments on the Quality of English Language
Moderate
Author Response
Reviewer 1: This is a narrative review about the treatment of appendiceal tumors.
The topic is interesting and almost completely exhaustive.
A moderate revision of the English language is needed.
The concepts reported by the authors are very well known and a section on future perspectives would be useful, including clinical implications of the treatment as well as of the disease.
It would also be useful to expand the references by updating them:
- doi: 10.1159/000536219
- https://doi.org/10.1007/978-3-031-36860-8_9
- doi: 10.7759/cureus.50783
Many others
Dear Reviewer, thank you for this insightful feedback. We appreciate your acknowledgment of the comprehensive overview of the background and rationale for our study. To address your suggestion, we have incorporated additional paragraph about future perspectives od diagnostics and treatment of appendiceal tumors (ATs) (lines 754-758) and provided additional paragraphs with suggested references (lines 344-349, 358-361)
“Future perspectives on the diagnosis and treatment of ATs should focus on systematizing the tools needed for the diagnosis of ATs, which, due to their rarity and nuanced nature, remain a challenge all the time. Treatment of ATs also remains a challenge, due to the inability to conduct large, randomized trials. However, with numerous original reports, we should aim to systematize guidelines for all ATs.”
“PMP can be classified into two distinct groups, namely unique methylation epigenotype (UME) and normal-like methylation epigenotype (NLME), determined by the methylation patterns in their promoters. The involvement of genes linked to neuronal development and synaptic signaling may contribute to the development of PMPs. [27] The most frequent genetics aberrations in PMP are similar to those In AMN and include KRAS, GNAS and TP53, ATM, ERBB2, FBXW7, NRAS and SMAD4 mutations. [27,28]”
“They are usually found accidentally, causing the symptoms of appendicitis. The occurrence of tumors leading to carcinoid syndrome is very rare and is associated with the spread of metastases. Immunohistochemical staining for synaptophysin, and chromogranin A is needed to confirm the neuroendocrine origin of NEN. [30]”
We trust that modifications contribute to the overall strength of our study.
Reviewer 2 Report
Comments and Suggestions for Authors
Congratulations for the comprehensive review. It is a well structured and interesting paper. I really appreciate the figures.
My only comment would be related to the treatment of PMP. Some cases of severe PMP, even after CRS and HIPEC, may require redo cases even with extensive intestine and colon resection followed with multivisceral transplant. (Reddy S et al. First Report With Medium-term Follow-up of Intestinal Transplantation for Advanced and Recurrent Nonresectable Pseudomyxoma Peritonei. Ann Surg. 2023 May 1;277(5):835-840. doi: 10.1097/SLA.0000000000005769. Epub 2022 Dec 5. PMID: 36468404; PMCID: PMC10082061.) I leave it for the Authors decision if they want to mention about this new approach for advanced PMP. I find it very interesting and review is a good format not only for the standardized approach but also for a discussion of potential upcoming strategies.
However, it does not change my recommendation for publication of the Authors' manuscript.
Author Response
Reviewer 2. Congratulations for the comprehensive review. It is a well structured and interesting paper. I really appreciate the figures.
My only comment would be related to the treatment of PMP. Some cases of severe PMP, even after CRS and HIPEC, may require redo cases even with extensive intestine and colon resection followed with multivisceral transplant. (Reddy S et al. First Report With Medium-term Follow-up of Intestinal Transplantation for Advanced and Recurrent Nonresectable Pseudomyxoma Peritonei. Ann Surg. 2023 May 1;277(5):835-840. doi: 10.1097/SLA.0000000000005769. Epub 2022 Dec 5. PMID: 36468404; PMCID: PMC10082061.) I leave it for the Authors decision if they want to mention about this new approach for advanced PMP. I find it very interesting, and review is a good format not only for the standardized approach but also for a discussion of potential upcoming strategies.
However, it does not change my recommendation for publication of the Authors' manuscript.
Dear Reviewer, thank you for this valuable observation and careful insight into our work. We have added the paragraph about the forementioned topic on PMP treatment which we believe will significantly improve the reception of this manuscript. (lines 523-529)
“A promising avenue for future research in the treatment of PMP seems to involve the refinement and exploration of radical exenteration and transplantation techniques. This is indicated by the observed enhancement in patient quality of life, despite the challenges posed by surgically related complications, leading to a 79% 1-year and 55% 5-year survival rate. The notable 91% rate of disease progression/recurrence in patients monitored for over 6 months underscores the imperative for additional investigation to optimize long-term efficacy and outcomes. [48]”
Your feedback has been invaluable in refining this aspect of our manuscript, and we appreciate your guidance.
Reviewer 3 Report
Comments and Suggestions for Authors
thank you for allowing me to review this recent review of the literature on the management of appendiceal tumors. the review is well written and informative. the decision-making algorithms are very useful. however, i do have a few comments and questions.
lines 61-63: I will list the different types of tumor in descending order of prognosis.
lines 75-76: if there is lymphoma on the surgical specimen, why should patients be operated on again?
lines 97-99: what were the exclusion criteria for the articles: any type of language?
line 272: the authors speak of endoscopic treatment of intra-appendicular lesions. what equipment for what benefit? isn't it more prudent to perform an appendectomy?
line 317: what should be done if a mucinous tumour of the appendix is suspected? exploratory laparoscopy; search for intraperitoneal lesion and calculation of ICP? I suggest that the authors add a paragraph on the management of appendicular lesions discovered on CT without emergency criteria.
Author Response
Reviewer 3: Thank you for allowing me to review this recent review of the literature on the management of appendiceal tumors. the review is well written and informative. the decision-making algorithms are very useful. however, i do have a few comments and questions.
lines 61-63: I will list the different types of tumor in descending order of prognosis.
lines 75-76: if there is lymphoma on the surgical specimen, why should patients be operated on again?
lines 97-99: what were the exclusion criteria for the articles: any type of language?
line 272: the authors speak of endoscopic treatment of intra-appendicular lesions. what equipment for what benefit? isn't it more prudent to perform an appendectomy?
line 317: what should be done if a mucinous tumour of the appendix is suspected? exploratory laparoscopy; search for intraperitoneal lesion and calculation of ICP? I suggest that the authors add a paragraph on the management of appendicular lesions discovered on CT without emergency criteria.
Dear Reviewer, thank you for your constructive feedback and insight into our work.
In the introduction, we decided to list ATs in terms of prevalence, due to the rarity of ATs diagnosis and diagnostic difficulties.
Paragraph in introduction on lymphomas has been revised (lines 81-83):
“However, if lymphoma or a larger NEN is identified, repeat surgery to obtain a radical surgical margin, cytoreductive surgery (CRS), or CTH might be required for more definitive treatment.”
A paragraph on exclusion criteria has been added (lines 116-117):
“The exclusion criteria encompassed publications in languages other than English and book chapters.”
The section on endoscopic treatment has been elaborated on (lines 505-509):
“Endoscopic ablation or minimally invasive appendectomy remain the recommended treatment for serrated lesions of the appendix due to their benign nature. This procedure is beneficial in the case of lesions found accidentally during endoscopic examination, providing histopathological diagnosis or a less traumatic alternative for the patient in the absence of signs indicating appendicitis.”
The paragraph on management depending on the CT result has been added to the Treatment section (501-504):
“Conducting a preoperative CT assessment of ATs proves advantageous by providing an approximate evaluation of mass size, identification of potential metastases, and sometimes determination of tumor type. This assessment aids in informed decision-making about the most suitable surgical approach for a given case. [39]”
Hopefully, these adjustments enhance the overall clarity and highlight the aspects of our research.